# Arm Circumference, Arm-to-Waist Ratio in Relation to Cardiovascular and All-Cause Mortality among Patients with Diabetes Mellitus

**DOI:** 10.3390/nu15040961

**Published:** 2023-02-15

**Authors:** Xinyu Xiao, Xinyi Yu, Huiping Zhu, Xiaobing Zhai, Shiyang Li, Wenzhi Ma, Meishou Ouyang, Keyang Liu, Ehab S. Eshak, Jinhong Cao

**Affiliations:** 1School of Public Health, Wuhan University, Wuhan 430071, China; 2School of Public Health, Capital Medical University, Beijing 100069, China; 3Center for Artificial Intelligence Driven Drug Discovery, Faculty of Applied Sciences, Macao Polytechnic University, Macau SAR, China; 4Public Health, Department of Social Medicine, Graduate School of Medicine, Osaka University, 2-2 Yamadaoka, Suita-shi 565-0871, Osaka, Japan; 5Department of Public Health, Faculty of Medicine, Minia University, Minia 61511, Egypt; 6Advanced Clinical Epidemiology, Medical data Science Unit, Public Health, Department of Social Medicine, Graduate School of Medicine, 2-2 Yamadaoka, Osaka University, Suita-shi 565-0871, Osaka, Japan; 7Public Health, School of Health, Calvin University, Grand Rapids, MI 49546, USA; 8School of Management, Hubei University of Traditional Chinese Medicine, Wuhan 430065, China; 9Research Center for the Development of Traditional Chinese Medicine, Hubei Province Project of Key Research Institute of Humanities and Social Sciences at Universities, Wuhan 430065, China

**Keywords:** arm circumference, waist circumference, arm-to-waist ratio, cardiovascular disease, mortality

## Abstract

Among patients with diabetes mellitus, limited studies have investigated the relationship between anthropometric parameters and cardiovascular disease (CVD), with all-cause mortality. We examined the associations of arm circumference (AC), arm-to-waist ratio (AC/WC), and CVD, with all-cause mortality among patients with diabetes. This is a cohort study of 5497 diabetic individuals aged 20 or over who were recruited in the National Health and Nutrition Examination Survey (NHANES) from 1999 to 2014. Cox proportional hazards regression models were used to analyze the associations between AC, AC/WC, and CVD, with all-cause mortality. We also conducted stratified analyses and explored the possible non-linear relation by restricted cubic splines. During a median follow-up of 7.2 years, there were 271 and 1093 cases of CVD and all-cause death. The multivariable adjusted hazard ratios (HRs) with 95% confidence intervals (Cis) of CVD mortality in Q2, Q3, and Q4 groups compared with Q1 group were 0.37 (0.22, 0.62), 0.24 (0.12, 0.48), 0.18 (0.07, 0.46) for AC, and 0.18 (0.07, 0.46), 0.34 (0.20, 0.60), 0.28 (0.15, 0.53) for AC/WC. Similar results were observed in the analysis for all-cause mortality risk. AC and AC/WC were both inversely associated with CVD and all-cause mortality among individuals with diabetes. It is important to pay attention to these anthropometric parameters of diabetic patients.

## 1. Introduction

Diabetes is becoming more common over the world, with high rates of cardiovascular disease (CVD) and all-cause death [1]. Approximately 422 million people worldwide have diabetes, most of whom live in low- and middle-income countries, and 1.5 million people die each year as a direct result of diabetes [2]. From 2015 to 2030, the global economic burden of adult diabetes will increase from $1.3 trillion [3] to $2.2 trillion and from 1.8% to a maximum of 2.2% of the global GDP [4]. The associations between body mass index (BMI), waist circumference (WC), waist-to-hip ratio, and diabetes have been widely reported and frequently reviewed in many studies [5,6]. Some anthropometric parameters are commonly used in diabetes screening [7] and the prediction of mortality [8].

The circumference of the upper arm measured at the midpoint between the tip of the shoulder and the tip of the elbow is known as arm circumference (AC). AC offers reliable clinical advantages: quick, portable, inexpensive, uncomplicated, noninvasive, and can be measured without difficulty [9]. It has been shown to predict subclinical atherosclerosis and other cardiometabolic illnesses in adults [10], as well as centripetal obesity and insulin resistance in diabetic patients [11], and it is an alternative anthropometric measure of obesity in type 2 diabetes and metabolic syndrome [12]. Some studies examined the relationships between AC and health outcomes. It seems that AC is associated with CVD [10] and all-cause mortality [13], though there exists a debate [14]. At the same time, WC was considered as a predictor of CVD risk and mortality. As an anthropometric indicator, waist circumference has unique strengths: it reflects the percentage of body fat in the abdomen [15], it is the best predictor of intra-abdominal fat changes during weight loss [16], it is a key element of the metabolic syndrome [17], it can predict diabetes [18], it is linked to the risk of cardiometabolic disease [19], and it can improve risk models for cardiovascular disease and other outcomes [20]. It is superior to BMI in predicting diabetes, as BMI fails to differentiate between skeletal muscle and body fat [21], which can have the opposite effect in diabetes patients. However, WC is still used less frequently than other anthropometric measurements in predicting the risk of CVD mortality because it is not a typical risk factor for cardiovascular mortality [18].

Few studies explored diabetic patients’ AC and AC/WC in relation to their mortality. Based on data from the National Health and Nutrition Examination Survey (NHANES), our study aimed to investigate the associations of AC and AC/WC with CVD and all-cause mortality among patients with diabetes. We hypothesized that larger AC and AC/WC would be associated with lower CVD and all-cause mortality risk in diabetic patients. If AC and AC/WC are proven to be predictive of CVD and all-cause mortality in our study, they can be used as simple and sensitive tools for the early detection of high-risk individuals among the diabetic population.

## 2. Materials and Methods

### 2.1. Study Population

NHANES is a nationwide survey of the non-institutionalized civilian population in the United States, designed to assess the health and nutrition status of adults and children. It was conducted by the National Center for Health Statistics (NCHS) at the Centers for Disease Control and Prevention (CDC) and collected demographic, laboratory, and examination data to determine risk factors and disease prevalence.

This study included participants aged 20 or over with diabetes from the NHANES 1999–2014 datasets. Participants with at least one of the following conditions were defined as having diabetes in NHANES: (1) self-reported doctor diagnosis of diabetes; (2) self-reported use of insulin or oral hypoglycemic medication; (3) fasting glucose ≥ 7.0 mmol/L; (4) glycated hemoglobin A1c (HbA1c) ≥ 6.5%. The study excluded participants who lacked information on AC, WC, mortality, and participants with a history of CVD at baseline. Finally, a total of 5497 participants remained in our cohort study (Figure 1).

### 2.2. Exposure Measurement

AC was measured when the participant stood upright with their arms hanging loosely. The measuring tape was wrapped around the right upper arm at a point that was perpendicular to the upper arm’s long axis. It was recorded to the nearest millimeter.

WC was measured directly against the skin at the superior lateral border of the iliac crests. The examiner stood on a participant’s right side, palpated the hip area to find the right ilium of the pelvis, and then drew a horizontal line above the most superior lateral boundary. After the individual took one normal breath, the examiner wrapped a steel tape around the waist at the measurement mark level, and the WC data were taken. The measurement was recorded to the nearest millimeter. More details are available on the website of NHANES (www.cdc.gov/nchs/nhanes/about_nhanes.htm (accessed on 12 April 2022)).

### 2.3. Covariate Assessment

A home interview was conducted during the baseline survey, followed by a physician examination at the mobile examination center (MEC). Self-reported data, such as age, gender, education, race/ethnicity, alcohol drinking status, smoking status, history of hypertension, history of cancer, and self-reported health were collected during the interview. Information on height and weight was collected in the MEC and body mass index (BMI) was calculated as weight (kg) divided by the square of height (m^2^). Laboratory data included total cholesterol (TC), HDL cholesterol (HDL-C), and fasting glucose. NHANES collected 24-h dietary recalls from each participant and estimated intakes of nutrients and other food components from those foods and beverages with the USDA’s Food and Nutrient Database for Dietary Studies (FNDDS) [22]. The FNDDS includes comprehensive information that can be used to code individual foods and portion sizes reported by participants and also includes nutrient values for calculating nutrient intake. The estimations of carbohydrate intake and dietary fiber were used in our study. In detail, age was classified as <40 years, 40–59, and ≥60 years. Race/ethnicity was categorized as Mexican American, non-Hispanic white, non-Hispanic black, or other. Education levels were categorized as less than high school, high school or equivalent, and college or above. Alcohol drinking status was classified as never drinking (0 g/day), moderate drinking (0.1 to 27.9 g/day for men and 0.1 to 13.9 g/day for women), and heavy drinking (≥28 g/day for men and ≥14 g/d for women) [23]. Participants were also grouped as never smoker, former smoker, and current smoker based on their responses to questions about smoking at least 100 cigarettes during their lifetime and whether they were currently smoking. Participants with HDL-C < 40 mg/dL, as well as TC, LDL-C, and triglyceride levels of ≥200, ≥130, and ≥130 mg/dL, respectively, were considered to be with dyslipidemia. History of hypertension and history of cancer were determined according to the answers of the question “Ever told by a doctor or other health professional that you had high blood pressure” and the question “Ever told by a doctor or other health professional that you had cancer or malignancy”. Self-reported health was classified as very good to excellent, good, and poor to fair.

### 2.4. Outcome Ascertainment

The National Center for Health Statistics created a linked mortality file that contains a probabilistic match between NHANES and National Death Index (NDI) information to determine mortality status. Deaths from CVD were defined as deaths caused by cardiac diseases (codes I00-I09, I11, I13, I20–I51) or cerebrovascular diseases (codes I60–I69) [24].

### 2.5. Statistical Analysis

In view of the complex, multistage, stratified, and cluster-sampling design of NHANES, all statistical analyses of our study used appropriate sample weights, strata, and primary sampling units linked to the NHANES data.

AC and AC/WC were both divided into quartiles (Q1–Q4). The characteristics of the study population were presented as mean ± standard deviation (SD) for continuous variables and numbers (percentages) for categorical variables. We used Cox proportional hazard models to explore the associations between AC, AC/WC, and CVD, with all-cause mortality. We adjusted for age in model 1. Model 2 was additionally adjusted for gender, race/ethnicity, education, alcohol drinking status, smoking status, and BMI. Model 3 was further adjusted for TC, HDL-C, carbohydrate intake, dietary fiber intake, and fasting glucose. Model 4 was a fully-adjusted model of history of hypertension, history of dyslipidemia, history of cancer, self-reported health, and all covariates in model 3.

In order to test whether these variables had an impact on the associations between the measured anthropometrics and the mortality risks, we performed stratified analyses by age (< or ≥60 years), gender (male or female), smoking status (non-smokers or smokers), alcohol drinking status (non-drinkers or drinkers), and BMI (non-obese: <30 kg/m^2^ or obese: ≥30 kg/m^2^). We also examined the non-liner relation between AC, AC/WC, and CVD, with all-cause mortality by multivariable Cox regression models based on restricted cubic splines. Data were analyzed using SAS 9.4 (SAS Institute, Cary, NC, USA), and two-sided *p* value < 0.05 was considered statistically significant.

## 3. Results

The characteristics of the study population are shown in Table 1. The elderly tended to have small ACs, while the middle-aged participants tended to have large ACs. Females made up the majority of the smallest AC group, whereas males made up the majority of the other three groups. Our study population was generally educated beyond high school and mostly non-Hispanic white in race. Additionally, larger AC groups showed higher BMI, carbohydrate intake, and lower HDL-C levels. Characteristics according to AC/WC are mostly consistent with AC, but those in the middle of AC/WC tend to have smaller BMIs, and larger AC/WC groups show higher HDL-C.

Table 2 showed the associations between AC, AC/WC, and CVD, with all-cause mortality. In comparison to Q1, all the other three quartiles had significantly lower risks of CVD and all-cause mortality. In the analysis of AC, HRs (95% CI) of diabetic patients with Q4 of AC were 0.36 (0.21, 0.64) for CVD mortality and 0.40 (0.30, 0.52) for all-cause mortality in the crude model. The inverse association between AC and the risk of CVD and all-cause mortality remained in models adjusted for various variables. In the fully-adjusted model, those with Q4 of AC had a significantly lower risk of CVD and all-cause mortality, with a HR (95% CI) of 0.19 (0.07, 0.48) and 0.29 (0.20, 0.43), respectively. With increasing AC, the risk of CVD and all-cause mortality decreased. In the analysis of AC/WC, the risk of CVD and all-cause mortality was also significantly lower in larger AC/WC groups; the fully-adjusted HR (95% CI) of Q4 was 0.28 (0.15, 0.53) for CVD mortality and was 0.43 (0.31, 0.62) for all-cause mortality.

Through stratified analysis in Figure 2, we found that in most subgroups, CVD and all-cause mortality still showed a decreasing trend with increasing AC or AC/WC. In addition, when AC changed from Q3 to Q4, the HR of CVD mortality increased in women [Q3: 0.16 (0.05, 0.51), Q4: 0.21 (0.06, 0.78)]. When AC/WC changed from Q3 to Q4, HRs for CVD and all-cause mortality increased in diabetic patients ≥ 60 years old [CVD: Q3: 0.33 (0.17, 0.64), Q4: 0.40 (0.19, 0.84); all-cause: Q3: 0.46 (0.32, 0.66), Q4: 0.47 (0.32, 0.68)]. No significant interaction was found in the stratified analysis (all *p* for interaction ≥ 0.05).

Figure 3 showed the relation between AC, AC/WC, and the HRs for CVD, with all-cause mortality. Both AC and AC/WC were inversely associated with CVD and all-cause death, independent of multiple demographic, socioeconomic, lifestyle, and dietary factors.

## 4. Discussion

In this cohort study, diabetic individuals with a larger AC showed a decreased risk of CVD and all-cause mortality after multi-variable adjustment, suggesting a larger AC might be advantageous for diabetic patients. A similar protective effect was also observed in the analysis of AC/WC when we took WC into consideration.

Previous studies have utilized AC to predict mortality risk. Studies conducted in children demonstrated that AC is as good or perhaps even better than other indicators in predicting mortality such as weight-for-height [25], age-specific BMI [26], and other common indicators [27]. A study of elderly people in Taiwan showed that AC was the strongest in predicting the risk of death at a follow-up of 12 months [28]. For Chinese older individuals residing in long-term care facilities, AC was a predictor of all-cause mortality, as participants with bigger ACs had lower all-cause mortality [29]. Mid-upper AC was also found to be more strongly related with 15-year mortality than BMI and calf circumference in both men and women in Netherlands [30]. However, a few studies reported contrary conclusions. A study in Chinese adults found the association between increased AC and higher risk of heart disease and atherosclerosis [10]; the findings of a South African study in children indicated that a higher risk of CVD was related to an increased AC [31]. On top of single anthropometric parameters like AC, a growing number of studies tend to use ratios to measure cardiovascular mortality risk, such as the waist-to-hip ratio (WHR), an important indicator for the diagnosis of central obesity [32]. Unlike previous studies, our study was conducted in diabetic patients to explore the associations between AC, AC/WC, and CVD, with all-cause mortality among them, considering people with diabetes are highly susceptible to co-morbidity with other illnesses [33]. We adjusted possible confounders including BMI to examine the relation between AC and health outcomes, finding that risks of CVD and all-cause mortality decreased with increasing AC. Furthermore, our findings revealed that the larger the AC/WC, the greater protective effect. AC/WC may be a good predictor of CVD and all-cause death in diabetic patients since a large AC/WC represents a healthy standard body size to some extent. If AC rises only as a result of obesity, the WC is more likely to rise as well, resulting in a lower or unchanged ratio.

The explanations behind the associations of AC, AC/WC, and health outcomes are still not very clear. On the one hand, exercise has been shown to have a preventive impact on CVD and diabetes [34,35], which may be reflected by the increase of AC and the decrease of WC. On the other hand, some studies showed that a larger AC indicated a higher muscle content, which was associated with a better quality of life and less possibility of falling down, and improved cardiovascular function and anti-aging [36]. Another study demonstrated that a larger AC may indicate a larger level of fat; excess fat can contribute to cardiovascular and metabolic disease, but a moderate level of fat can protect against incidental injury, which is especially significant in older persons [37]. At the same time, WC was seen as a sensitive indicator used to screen for diabetes and was even more relevant than weight and BMI [38], and the linear association between WC and all-cause mortality was found in many studies [39]. Therefore, evaluating AC and AC/WC may be useful to predict CVD and all-cause mortality risk, especially in diabetic patients.

We found that as AC increased to a high level among female patients, the risk of CVD mortality seemed to increase. On the one hand, some studies have already reported that the higher the arm-leg ratio, the higher the prevalence of the CVD risk factor, which is more pronounced in women [40], suggesting the increase of CVD risk may be due to the increase in cardiovascular risk factors caused by the increase of fat in women. On the other hand, diabetes leads to higher levels of inflammatory markers and higher rates of nitric oxide release in women than in men, resulting in reduced protective effects of estrogen on body fat distribution, insulin action, and more impaired endothelial function [41,42]. For female diabetic patients, a high level of ACs is more likely to reflect the over-increase of fat, which adds to the cardiovascular burden. Additionally, the over-increase of fat can directly increase cardiovascular disease morbidity and mortality, mediated by obesity-induced structural and functional adaptations of the cardiovascular system to accommodate overweight, as well as the effects of adipokines on inflammation and vascular homeostasis, leading to a pro-inflammatory and pro-thrombotic environment [43].

Previous studies have shown that in older adults, increased AC levels are associated with reduced mortality risk [44,45,46] and reduced WC is associated with increased mortality [47]. However, when AC/WC increased from Q3 to Q4, the risk of CVD and all-cause mortality increased among diabetic patients ≥60 years in our study. The obesity paradox may provide an explanation for this finding: Some studies found that weight loss rather than weight gain is associated with increased morbidity and mortality during follow-up [48]. Overweight individuals with multiple chronic diseases, especially the elderly, may have shown lower all-cause mortality and cardiovascular mortality compared to patients of normal weight [49], and an overweight condition in an elderly person seems to prevent death [50]. The same is true for overweight diabetic patients. The decrease of AC/WC from Q3 to Q4 among diabetic patients ≥60 years may reflect unfavorable weight loss, which increases the risk of CVD and all-cause mortality.

In our study, we focused on diabetic patients and found an inverse association between AC, AC/WC, and CVD, with all-cause mortality among them. In comparison to AC/WC, AC showed a better dose-response relationship with the risk of CVD and all-cause mortality. AC is easy to measure, non-invasive, and shows inverse association with primary health outcomes of diabetic patients significantly. In addition, the inverse association still remains in most subgroups according to our results, suggesting the potential value of being widely used. Therefore, it may be a good predictor for CVD and all-cause mortality among diabetic patients. Our findings emphasize the importance of an AC measurement as a predictor of mortality for diabetic patients, which helps to identify individuals early who are at a high risk of CVD and all-cause mortality and precisely implement prevention. More studies are needed to further confirm the relation and clarify the possible mechanisms.

The strengths of our study are as follows: first, it has a large, nationally representative sample and long-term follow-up. Second, this study was the first to examine the associations of AC, AC/WC, and CVD, with all-cause mortality among patients with diabetes. Our research has several limitations as well: first, the data of ACs and WCs were gathered at the time of baseline, and diabetic patients may have varied physical features at different phases, so the AC or WC we acquired may not reflect the long-term stable status of patients. Second, although we have adjusted for lots of potential confounders, there may be additional confounding due to uncollected data. AC may be influenced by many diseases in relation to the risk of mortality. A lower AC, which is more than an independent predictor of mortality, may be the result of increased mortality risk from the disease itself.

## 5. Conclusions

Based on this cohort study, a larger AC and AC/WC were associated with lower CVD and all-cause mortality in participants with diabetes, suggesting that AC and AC/WC may be possible predicting indicators for CVD and all-cause mortality among diabetic patients. It is important to confirm this association and further explore the potential meanings of these anthropometric parameters among diabetic patients in the future. Our findings underlined the importance of using anthropometric assessment tools like AC and AC/WC to evaluate the risk of CVD and all-cause mortality in the population with diabetes.

## Figures and Tables

**Figure 1 nutrients-15-00961-f001:**
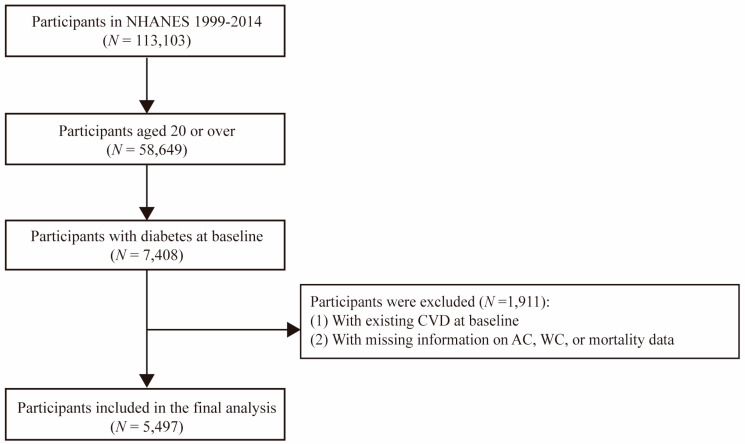
Flowchart of participants with diabetes included in the analysis.

**Figure 2 nutrients-15-00961-f002:**
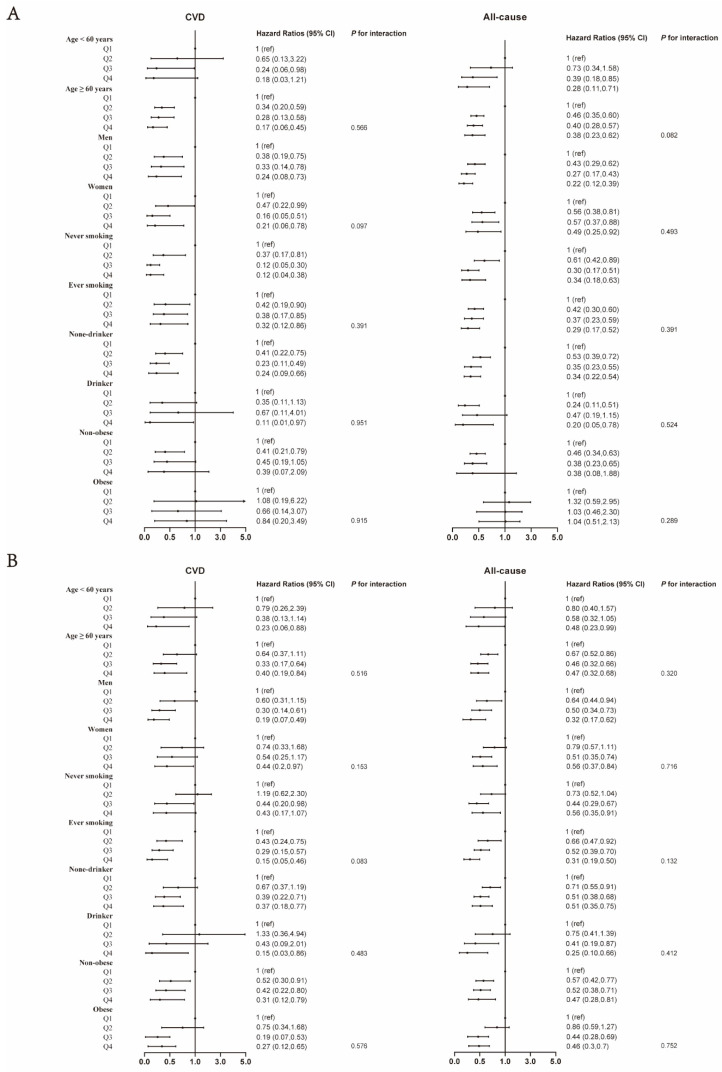
Associations between AC (**A**), AC/WC (**B**), and risk of CVD, with all-cause mortality stratified by age, gender, alcohol drinking status, smoking status, history of hypertension. HRs were adjusted for age (not adjusted in subgroup analysis by age), gender (not adjusted in subgroup analysis by gender), race/ethnicity, education, alcohol drinking (not adjusted in subgroup analysis by alcohol drinking), alcohol drinking (not adjusted in subgroup analysis by alcohol drinking), BMI (not adjusted in subgroup analysis by BMI), total cholesterol, high-density lipoprotein cholesterol, carbohydrate intake, dietary fiber intake, fasting glucose, history of hypertension, and history of dyslipidemia, history of cancer, self-reported health.

**Figure 3 nutrients-15-00961-f003:**
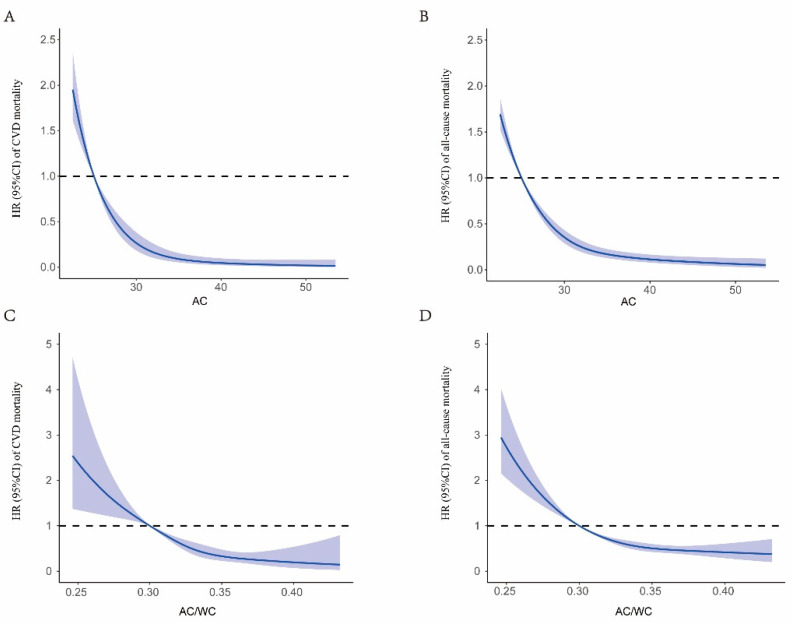
Dose-response association between AC and CVD mortality risk (**A**), AC and all-cause mortality risk (**B**), AC/WC and CVD mortality risk (**C**), AC/WC all-cause mortality risk (**D**). Associations were examined by multivariable Cox regression models based on restricted cubic splines. The solid line represents estimates of hazard ratios and the dashed line represents 95% CIs. HRs were adjusted for age, gender, race/ethnicity, education, alcohol drinking status, smoking status, BMI, total cholesterol, high-density lipoprotein cholesterol, carbohydrate intake, dietary fiber intake, fasting glucose, history of hypertension, and history of dyslipidemia, history of cancer, self-reported health.

**Table 1 nutrients-15-00961-t001:** Baseline characteristics of the study population, according to AC and AC/WC.

Characteristic	AC				AC/WC			
	Q1 <31.40	Q2 31.40–34.49	Q3 34.50–37.99	Q4 ≥38.00	Q1 <0.30	Q2 0.30–0.32	Q3 0.33–0.34	Q4 ≥0.35
**Number of participants**	1351	1385	1341	1420	1374	1375	1373	1375
**Age, years**								
<40	85 (9.70)	96 (9.10)	142 (15.06)	207 (16.17)	83 (8.32)	116 (11.24)	136 (12.38)	195 (19.27)
40–59	334 (33.93)	451 (40.94)	518 (45.14)	656 (56.67)	331 (34.49)	441 (40.46)	564 (51.28)	623 (53.93)
≥60	932 (56.36)	838 (49.96)	681 (39.80)	557 (27.16)	960 (57.19)	818 (48.30)	673 (36.34)	557 (26.80)
**Gender**								
Male	641 (42.34)	746 (53.16)	743 (58.04)	680 (50.42)	762 (54.65)	749 (56.16)	740 (53.37)	559 (41.54)
Female	710 (57.66)	639 (46.84)	598 (41.96)	740 (49.58)	612 (45.35)	626 (43.84)	633 (46.63)	816 (58.46)
**Race/ethnicity**								
Mexican American	320 (9.53)	362 (10.86)	300 (8.65)	191 (6.15)	274 (7.58)	348 (10.01)	306 (9.43)	245 (7.40)
Non-Hispanic white	465 (55.32)	478 (58.81)	558 (68.38)	522 (63.30)	683 (73.75)	538 (65.73)	424 (55.44)	378 (53.37)
Non-Hispanic black	240 (10.76)	312 (13.47)	330 (13.36)	563 (21.61)	236 (9.12)	282 (10.91)	370 (15.54)	557 (25.10)
Other	326 (24.39)	233 (16.86)	153 (9.61)	144 (8.94)	181 (9.55)	207 (13.35)	273 (19.59)	195 (14.12)
**Education**								
Less than high school	590 (30.35)	566 (28.11)	460 (21.34)	417 (20.39)	563 (27.73)	534 (26.03)	485 (23.02)	451 (21.52)
High school or equivalent	301 (24.55)	278 (22.44)	309 (24.17)	378 (29.83)	317 (27.14)	292 (24.78)	322 (23.88)	335 (26.30)
College or above	456 (45.11)	538 (49.45)	570 (54.49)	625 (49.77)	491 (45.12)	546 (49.19)	564 (53.10)	588 (52.18)
**Alcohol drinking status**								
Never drinking	1050 (78.91)	1078 (80.15)	1082 (80.93)	1170 (81.66)	1118 (83.24)	1079 (79.85)	1086 (79.42)	1097 (79.84)
Moderate drinking	112 (10.64)	138 (10.50)	97 (9.13)	100 (10.01)	104 (10.16)	113 (9.09)	116 (10.73)	114 (10.10)
Heavy drinking	107 (10.45)	104 (9.35)	116 (9.94)	108 (8.33)	80 (6.60)	118 (11.06)	124 (9.85)	113 (10.05)
**Smoking status**								
Never smoker	660 (47.36)	691 (49.13)	644 (49.51)	755 (54.66)	607 (42.74)	658 (49.48)	680 (48.72)	805 (60.51)
Former smoker	440 (33.74)	458 (34.14)	469 (33.36)	424 (27.67)	517 (36.74)	488 (34.79)	429 (32.59)	357 (23.98)
Current smoker	251 (18.90)	234 (16.73)	224 (17.13)	240 (17.68)	250 (20.52)	227 (15.73)	260 (18.68)	212 (15.50)
**BMI, kg/m^2^**	25.02 ± 3.08	29.23 ± 3.29	32.75 ± 3.60	40.50 ± 6.40	32.10 ± 7.32	31.47 ± 7.02	31.83 ± 6.83	32.51 ± 7.50
**Total cholesterol, mg/dL**	199.45 ± 52.37	197.05 ± 45.20	197.93 ± 52.40	191.67 ± 44.82	191.78 ± 49.49	196.88 ± 48.44	199.45 ± 50.48	197.77 ± 46.56
**HDL Cholesterol, mg/dL**	52.61 ± 15.51	48.40 ± 13.40	46.28 ± 12.80	45.45 ± 12.08	47.21 ± 13.42	47.29 ± 13.77	48.20 ± 13.90	49.94 ± 13.80
**Carbohydrate intake, gm**	213.06 ± 106.94	217.14 ± 102.17	233.19 ± 113.55	235.79 ± 125.54	218.27 ± 110.59	229.98 ± 110.64	230.70 ± 119.61	220.94 ± 110.38
**Dietary fiber intake, gm**	16.32 ± 9.96	16.19 ± 10.42	17.01 ± 11.11	15.59 ± 10.21	15.79 ± 9.91	16.59 ± 10.02	16.49 ± 10.47	16.19 ± 11.29
**Fasting Glucose, mmol/L**	8.93 ± 3.79	8.89 ± 3.80	9.11 ± 3.60	8.62 ± 3.09	8.99 ± 3.77	8.75 ± 3.43	8.93 ± 3.52	8.88 ± 3.62
**History of hypertension**								
Yes	733 (48.57)	786 (55.39)	832 (57.51)	982 (66.93)	896 (67.91)	806 (54.80)	818 (56.32)	813 (53.11)
No	617 (51.43)	594 (44.61)	504 (42.49)	433 (33.07)	473 (32.09)	564 (45.20)	552 (43.68)	559 (46.89)
**History of dyslipidemia**								
Yes	892 (64.75)	956 (68.95)	973 (74.53)	993 (71.71)	959 (71.44)	990 (73.97)	959 (69.85)	906 (66.22)
No	459 (35.25)	429 (31.05)	368 (25.47)	427 (28.29)	415 (28.56)	385 (26.03)	414 (30.15)	469 (33.78)
**History of cancer**								
Yes	189 (16.91)	183 (15.09)	154 (13.97)	156 (11.79)	221 (19.95)	184 (14.56)	160 (13.88)	117 (8.78)
No	1161 (83.09)	1197 (84.91)	1186 (86.03)	1262 (88.21)	1152 (80.05)	1188 (85.44)	1210 (86.12)	1256 (91.22)
**Self-reported health**								
Very good to excellent	238 (25.84)	281 (30.42)	270 (26.6)	169 (14.72)	223 (21.13)	229 (23.28)	244 (23.8)	262 (27.14)
Good	448 (40.05)	464 (41)	487 (45.53)	497 (46.57)	477 (43.36)	480 (42.52)	479 (43.17)	460 (45.54)
Poor to fair	463 (34.11)	431 (28.58)	386 (27.87)	559 (38.7)	498 (35.51)	472 (34.2)	455 (33.03)	414 (27.32)

Values are mean ± SD for continuous variables or numbers (percentages) for categorical variables.

**Table 2 nutrients-15-00961-t002:** Hazard ratios (95% CIs) of mortality with AC and AC/WC.

Characteristic	AC				AC/WC			
	Q1 <31.40	Q2 31.40–34.49	Q3 34.50–37.99	Q4 ≥38.00	Q1 <0.30	Q2 0.30–0.32	Q3 0.33–0.34	Q4 ≥0.35
CVD								
Number of participants	1351	1385	1341	1420	1374	1375	1373	1375
Deaths/person-years	121/9510	64/10,812	46/10,753	40/10,986	122/9459	75/10,306	44/10,664	30/11,631
Unadjusted	1[Reference]	0.52 (0.29, 0.94)	0.40 (0.23, 0.70)	0.36 (0.21, 0.64)	1[Reference]	0.59 (0.34, 1.02)	0.28 (0.16, 0.49)	0.17 (0.09, 0.33)
Model 1	1[Reference]	0.51 (0.29, 0.87)	0.46 (0.27, 0.76)	0.52 (0.31, 0.86)	1[Reference]	0.62 (0.37, 1.05)	0.34 (0.19, 0.59)	0.25 (0.13, 0.47)
Model 2	1[Reference]	0.34 (0.20, 0.57)	0.23 (0.12, 0.44)	0.17 (0.07, 0.45)	1[Reference]	0.62 (0.37, 1.05)	0.33 (0.19, 0.58)	0.26 (0.14, 0.51)
Model 3	1[Reference]	0.34 (0.20, 0.57)	0.24 (0.13, 0.45)	0.17 (0.07, 0.45)	1[Reference]	0.61 (0.37, 0.99)	0.33 (0.19, 0.58)	0.25 (0.13, 0.49)
Model 4	1[Reference]	0.37 (0.22, 0.62)	0.24 (0.12, 0.48)	0.19 (0.07, 0.48)	1[Reference]	0.65 (0.38, 1.09)	0.34 (0.20, 0.60)	0.28 (0.15, 0.53)
All-cause mortality								
Number of participants	1351	1385	1341	1420	1374	1375	1373	1375
Deaths/person-years	402/9510	288/10,812	217/10,753	186/10,986	437/9459	283/10,306	208/10,664	165/11,631
Unadjusted	1[Reference]	0.61 (0.45, 0.81)	0.46 (0.34, 0.63)	0.40 (0.30, 0.52)	1[Reference]	0.63 (0.48, 0.82)	0.39 (0.28, 0.55)	0.27 (0.19, 0.39)
Model 1	1[Reference]	0.60 (0.46, 0.79)	0.52 (0.39, 0.69)	0.55 (0.43, 0.72)	1[Reference]	0.67 (0.53, 0.85)	0.48 (0.35, 0.65)	0.39 (0.28, 0.54)
Model 2	1[Reference]	0.47 (0.36, 0.62)	0.34 (0.24, 0.48)	0.29 (0.19, 0.43)	1[Reference]	0.69 (0.53, 0.88)	0.48 (0.36, 0.65)	0.43 (0.30, 0.60)
Model 3	1[Reference]	0.46 (0.35, 0.61)	0.35 (0.25, 0.49)	0.29 (0.20, 0.44)	1[Reference]	0.67 (0.52, 0.86)	0.48 (0.36, 0.64)	0.42 (0.30, 0.59)
Model 4	1[Reference]	0.47 (0.36, 0.61)	0.35 (0.25, 0.49)	0.29 (0.20, 0.43)	1[Reference]	0.69 (0.53, 0.88)	0.49 (0.37, 0.64)	0.43 (0.31, 0.62)

Model 1 was adjusted for age. Model 2 was further adjusted for gender, race/ethnicity, education, alcohol drinking status, smoking status, BMI. Model 3 was further adjusted for total cholesterol, high-density lipoprotein cholesterol, carbohydrate intake, dietary fiber intake, fasting glucose. Model 4 was additionally adjusted for history of hypertension, history of dyslipidemia, history of cancer, self-reported health.

## Data Availability

The datasets generated and analyzed during the current study are available in the National Health and Nutrition Examination Survey, https://www.cdc.gov/nchs/nhanes/ (accessed on 12 April 2022).

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
