# Peer review of "Arm Circumference, Arm-to-Waist Ratio in Relation to Cardiovascular and All-Cause Mortality among Patients with Diabetes Mellitus"

_nutrients, 2023, doi:10.3390/nu15040961_

Round 1
Reviewer 1 Report
Comments
The present study used arm circumference and arm-to-waist ratio to predict the risk of CVD and all cause mortality. The idea is new and the writing is easy to understand.
However, several points need to be clarified:
1) Page 2, LDL cholesterol could write as LDL-C shorted version since the whole document used it several times.
2) On page 3, the category of alcohol consumption should list the reference documents.
3) On page 3, line116-117, serum lipids have tight multicollinearity, normally, we did not adjust them together. Please check, chose some of them for adjusting. And select key confounder for adjustment.
4) what effects of stratified analysis results could help in adjusting results, please add to the discussion.
5) How did the author calculate the amount of carbohydrate intake and dietary fiber intake, individually?
6) please adjust the Y axis to the same range for A &B, C&D, in figure 3.
7) please write several sentences to clarify why you think AC is a good predictor in diabetes patients.
8) It is very difficult to understand why larger AC/WC protects against CVD in those experienced smoking. Could authors explain more about this phenomenon?
Author Response
Thank you for your comments. The comments are all valuable and helpful for revising and improving our paper. We have taken all these comments into account and made correction accordingly. Please see the attachment.

Reviewer 2 Report
The present study is clear and well written, and its topic is interesting.
Main limitations are the single measurement of anthropometric variables and the many potential confounders that may impact on mortality. A lower AC, indeed, rather than an independent predictor of mortality, may be the consequence of a disease increasing by itself the risk of death.
An index of health status (for example, Charlson index) should be added in the model to take into account the presence of comorbidities, which heavily impact on survival.
Author Response

(The authors gave the same response as above.)

Reviewer 3 Report
Thank you for the opportunity to comment on this document. The subject is interesting and requires special vigilance. With that being said, I do have a few concerns. 1º I would like to know what is the novelty of the article and its contribution to the knowledge of the scientific community. 2º The introduction should be expanded to provide a solid background of the main idea of the study. Also, authors should end the introduction with a clear hypothesis.
Author Response

(The authors gave the same response as above.)

Round 2
Reviewer 2 Report
I have no further suggestions